# Amino Acid Absorption Profiles in Growing Pigs Fed Different Protein Sources

**DOI:** 10.3390/ani11061740

**Published:** 2021-06-10

**Authors:** Jan V. Nørgaard, Iulia C. Florescu, Uffe Krogh, Tina Skau Nielsen

**Affiliations:** Department of Animal Science, Aarhus University, 8830 Tjele, Denmark; iulia.florescu@skretting.com (I.C.F.); uffekrogh@anis.au.dk (U.K.); TinaS.Nielsen@anis.au.dk (T.S.N.)

**Keywords:** amino acid absorption, metabolism, pigs, protein

## Abstract

**Simple Summary:**

The speed by which amino acids are absorbed into the blood after intake of different protein sources may affect their metabolism and utilization. A better understanding of the absorption pattern can be used to optimize the formulation of diets for pigs and to reduce the nitrogen excretion to the environment. We studied the amino acid appearance in blood of growing pigs after a meal, as influenced by protein source (wheat, soybean meal, enzyme-treated soybean meal, hydrothermally-treated rapeseed meal, casein, or hydrolyzed casein). The amino acid concentration in plasma was influenced by both time after feeding and the protein source. Overall, the highest concentrations were found at 60 min after feeding for all diets, and soybean meal had a prolonged AA absorption compared to especially casein and hydrolyzed casein. We conclude that the AA absorption profiles did not indicate clear differences among protein sources, allowing categorizing in fast and slow proteins sources, but the results show differences in the duration of AA absorption.

**Abstract:**

The aim of the present study was to determine postprandial amino acid (AA) appearance in the blood of growing pigs as influenced by protein source. Seven growing pigs (average body weight 18 kg), in a 7 × 5 Youden square design, were fitted with a jugular vein catheter and fed seven diets containing wheat, soybean meal, enzyme-treated soybean meal, hydrothermally-treated rapeseed meal, casein, hydrolyzed casein, and a crystalline AA blend with the same AA profile as casein. The latter was not eaten by the pigs, therefore being excluded. Blood samples were collected at −30, 30, 60, 90, 120, 180, and 360 min after a meal and analyzed for free AA. Overall, plasma AA concentrations were highest 60 min after feeding. There were no differences in plasma AA concentration between casein and hydrolyzed casein, but soybean meal resulted in lower AA plasma concentrations compared with enzyme-treated soybean meal at 60 and 120 min after feeding. There were no differences between hydrothermally-treated rapeseed meal and soybean meal. In conclusion, the ingredients could not clearly be categorized as being slow or fast protein with regard to protein digestion and absorption of AA, but soybean meal resulted in a prolonged appearance of plasma AA compared to casein and hydrolyzed casein.

## 1. Introduction

It is crucial for optimal growth in pigs that all the essential amino acids (AA) are available in sufficient amounts and proportions in the diet, which also minimizes nitrogen excretion to the environment [1]. When optimizing pig diets, the minimum AA recommendations are fulfilled by including different protein sources, as well as crystalline AA, with major focus on the feedstuffs AA concentration and standardized ileal AA digestibility. 

The timing of occurrence of AA in plasma may be different among protein sources, and this will likely affect the utilization of AA. In human nutrition, the postprandial AA appearance in plasma has been categorized as a fast appearance in high concentrations followed by a rapid decrease (fast protein) or as a lower concentration with a slower clearance rate (slow protein), as described by Boirie et al. [2]. Moreover, from human studies, it has been shown that casein may result in a lower AA concentration in plasma compared with hydrolyzed casein or whey [3,4], although Horner et al. [5] found no significant differences on the postprandial AA concentrations between intact and hydrolyzed casein. 

The protein structure may affect the protein digestion and AA absorption kinetics due to enzyme affinity and transporter specificity [6]. Soluble proteins and free AA are absorbed faster than protein bound AA, and this can lead to AA catabolism and oxidation [7]. The transport across the intestinal membrane is faster for small peptides than for free AA because the competition for transporters is higher for free AA compared with unspecific peptide transporters [8]. The AA absorption depends on transport mechanisms across the intestinal wall where a competition for transport can be formed depending on the protein source and intake [9]. Further, considerable metabolism of some AA takes place in intestinal enterocytes [10], as well as bacterial activity utilizing AA in the intestinal lumen [11]. 

The AA appearance in portal blood is also dependent on the type of carbohydrates present in the diet. Indeed, lactose versus maize starch [12], maltose-dextrin added to a hydrolyzed casein diet [8], and the type of starch (resistant or non-resistant [13]) have been shown to affect AA absorption in pigs. 

In the present study, we hypothesized that crystalline AA and peptides result in a faster appearance of AA in plasma compared with intact dietary protein, and that origin and processing of protein sources will lead to differences in their digestion and absorption kinetics. The objective of the study was to evaluate the effect of different types of protein sources on postprandial AA appearance in plasma in pigs fed commercially used protein sources, as well as crystalline AA, casein, and hydrolyzed casein, as the sole sources of AA. 

## 2. Materials and Methods

### 2.1. Animals and Diets

Seven cross-bred ((Danish Landrace × Yorkshire) × Duroc) pigs with an initial average BW of 18 ± 1.25 kg were used. The pigs were housed individually, the temperature was maintained at 24 °C and humidity was maintained at 60%. The evaluated sources of protein were wheat, soybean meal (SBM), enzyme-treated soybean meal (HP 300, Hamlet Protein A/S, Horsens, Denmark), hydrothermally-treated rape seed meal (Lewistar, HaGe Nord AG, Kiel, Germany), casein (DLG a.m.b.a., Copenhagen, Denmark), hydrolyzed casein, and crystalline AA (C-AA). These sources of protein was chosen because they are the most abundantly used under Danish production conditions. Enzyme-treated soybean meal was produced from the same batch as the soybean meal used in this experiment. Hydrolyzed casein was prepared by enzyme hydrolysis of casein following the method described by Deglaire et al. [14]. In short, casein was dissolved by continuous mixing in deionized water (1:10, wt:vol) overnight. The obtained solution was then heated to 50 °C, and pancreatin was added in an enzyme to substrate ratio of 0.0089 at pH 8.0 for 1 h and 50 min at 50 °C. Then, the enzyme was inactivated at 85 °C for 20 min, and the solution was freeze-dried. The crystalline AA diet was based on the AA profile of casein with glutamic acid and proline used to substitute the crude protein (CP) contribution of the AA not available as crystalline AA (Cys, Arg, Asp, Ser). The experimental diets were formulated with the aim of being iso-nitrogenous (200 g CP/kg as-fed basis) by diluting the protein sources with nitrogen-free ingredients [15], except for wheat, which contained less than 200 g CP/kg (as-fed) and, therefore, fed without dilution. The ingredient composition of the seven experimental diets is presented in Table 1, the AA content of the protein sources in Table 2, and the analyzed chemical composition of the diets is presented in Table 3.

### 2.2. Protocol

The animal experimental procedures were carried out in accordance with the Danish Ministry of Justice, Law no. 253/08, March 2013 concerning animal experiments and care, and license issued by the Danish Animal Experiments Inspectorate, Ministry of Food, Agriculture and Fisheries, the Danish Veterinary and Food Administration. 

The seven pigs were fitted with a jugular vein catheter three days before the experimental period. Starting with day one of the experiment, the pigs were fed the experimental diets in the morning, amounting to 25**–**35 g/kg BW^0.75^ to assure the feed was completely eaten. The experimental diets were provided in a meal form and mixed with nitrogen-free apple juice in a 1:1 ratio to enhance taste and reduce the time for consuming the diets. In the afternoon (after sampling), the pigs had ad libitum access to a commercial growing pig diet for one hour to support a minimum intake of essential nutrients to support health and basic metabolism. The pigs had free access to water.

The experiment was designed as a randomized 7 × 5 Youden square design where the seven pigs were randomly fed one of the seven diets for one day during five days, so that no pig received the same diet more than once. On the first two experimental days, pigs were provided 25 g feed/kg BW^0.75^. However, from experimental day three, it was decided to increase the amount of feed per meal to 35 g/kg BW^0.75^ since pigs were able to consume more feed than expected and in order to maximize the AA response in blood. The effect of this change in feed allowance was balanced for all diets and included in the random effect of day in the statistical analysis of data. We have previously validated that 1.5 day of experimental feeding was sufficient in branched chain AA dose-response studies to reflect diet specific changes in the plasma AA profile following feeding a new diet [16]. The pigs received the experimental diet at 0730 h every morning, and the feed was removed 15 min later to avoid potential variation in AA profile in blood caused by prolonged feed intake. Blood samples (1 mL per time point) were collected at 30 min before and 30, 60, 90, 120, 180, and 360 min after the morning meal through the permanent jugular vein catheter. Blood samples (lithium heparin tubes; Greiner Bio-One GmbH, Kremsmünster, Austria) were centrifuged at 5 °C for 20 min at 3500 rpm (3000 *g*) and plasma stored at **−**80 °C until AA analysis. 

### 2.3. Chemical Analysis

Each protein source and diet was analyzed for AA composition and CP (European Commission, 2009). In short, for AA analysis, the samples were hydrolyzed at 110 °C for 23 h with performic acid oxidation for Cys and Met and without oxidation for all the other AA. Amino acids were quantified by photometric detection with ninhydrin reaction after separation by ion exchange chromatography (Biochrome 30+ Amino Acid Analyzer; Biochrome, Cambridge, UK) and calibrated with standards for acidic, neutral and basic AA (Sigma Aldrich, St. Louis, MO, USA). Heparinized plasma AA were quantified as described by Larsen et al. [17] using an isotope dilution method with AA analysis by GC-MS with *N*-tert-butyldimethyl AA derivatization.

### 2.4. Data Analysis 

Most pigs refused to eat the C-AA diet, probably due to poor palatability. Therefore, this diet was excluded from the statistical evaluation. Data were analyzed by the MIXED procedure of SAS (version 9.4; SAS Inst., Inc., Cary, NC, USA) using a model including time (−30, 30, 60, 90, 180, 360) and diet (wheat, soybean meal, enzyme-treated soybean meal, hydrothermally-treated rapeseed meal, casein, hydrolyzed casein) and their interactions as fixed effects. The pig and day number were included as random effects, and a first order covariance structure was used to account for the correlation between repeated measures within pigs on each sampling day. Plasma AA concentration at time −30 min was included as a covariate to account for the initial variation in plasma AA concentration. Statistical significance was accepted at *p* < 0.05. Data in the tables and figures represent least squares means (LSM) and standard error of the mean (SEM). Contrast statements were used to determine differences between protein sources at each sampling time. The area under curve (AUC), determined as measure for AA absorption and transmission to the systemic bloodstream, was calculated using the trapezoidal method with the MODEL procedure of SAS where AUC was obtained for each individual pig in time intervals. 

## 3. Results

The animals stayed healthy throughout the experimental period. The results shown in Figure 1, Figure 2 and Figure 3 are presented as both plasma AA concentrations (a) and as plasma AA concentrations in proportion to the maximum concentration (b), the latter to visualize the kinetics of AA absorption. The postprandial response in total AA concentration in plasma peaked 60 min after feeding for all diets, but there was an interaction between time and treatment (*p* < 0.001; Figure 1a). 

There was no difference between the total AA concentration before and after feeding for the wheat diet (*p* > 0.10). There were no differences in total AA concentration between wheat, soybean meal or hydrothermally-treated rapeseed meal 30 min after feeding (*p* > 0.10), but the AA concentration was higher (*p* < 0.05) for soybean meal and hydrothermally-treated rapeseed meal compared with wheat 180 min after feeding. At 180 min after feeding, the total AA concentration was higher (*p* < 0.05) for hydrolyzed casein than for casein. In contrast, the total AA concentration was higher (*p* < 0.05) for casein than for hydrolyzed casein at 360 min after feeding. The AA concentration in plasma following the casein, hydrolyzed casein and enzyme-treated soybean meal diet was higher (*p* < 0.05) compared with wheat, hydrothermally-treated rapeseed meal and soybean meal at all time-points, except at 360 min post-feeding (no difference). Soybean meal and enzyme-treated soybean meal showed different total AA concentration at 60 and 120 min after feeding, with enzyme-treated soybean meal resulting in a higher AA concentration (*p* < 0.05), but there were no difference between the two treatments after 120 min.

The soybean meal diet and the H-Cas diet represented two different types of responses (Figure 1b); a fast increase and a slow clearance of total aa and a fast increase with a fast clearance, respectively, whereas the other protein sources represented a response in between. The major differences among the diets were observed after 360 min. The highest increase rate (calculated as percentage increase from time point 0 min to the time point with the highest concentration) and corresponding decrease rate were observed for casein and hydrolyzed casein. The soybean meal diet resulted in the steadiest concentration of total AA. 

The concentration of essential AA in plasma (Figure 2a) followed the patterns of total AA concentrations. Pigs fed the soybean meal diet had a slow but continuous increase in essential AA with the maximum concentrations at 360 min postprandial (Figure 2b).

Non-essential AA’s had slightly different absorption patterns compared to the essential AA´s. The casein and hydrolyzed casein diets resulted in the highest essential AA concentrations and wheat in the lowest concentrations (Figure 3a). The soybean meal diet resulted in a steady absorption from 60 min after feeding (Figure 3b), while hydrolyzed casein had the greatest clearance of non-essential AA at 360 min. Hydrothermally-treated rapeseed meal also had a prolonged plasma clearance, like soybean meal. 

During all time intervals, the highest accumulated AA appearance in blood (area under the curve) was obtained for hydrolyzed casein and casein and the lowest for wheat (Table 4). The two soy products tended to be different with enzyme-treated soybean having a higher AA appearance in blood compared with soybean meal.

## 4. Discussion

The AA concentration in the plasma is the result of several factors of which the main effects are the AA profile and concentration in the diet, time after feeding, protein digestibility, AA absorption, and metabolism in the small intestine and liver [10,18]. There may also be a contribution of AA to the venous blood by AA mobilized from body reserves, especially in a fasting phase. Moreover, the appearance of AA in blood depends on the concentrations of other AA as shown in AA dose-response studies where supplementation of the first limiting AA reduces the concentration of other non-limiting AA (e.g., Reference [19]). The feed evaluation systems used to assess the availability of energy and nutrients in the diets for pigs, take into account the ileal digestibility of protein and essential AA. Accordingly, the content of protein, energy and the profile of AA in a given diet can be related to the estimated requirements of pigs in a given physiological state. However, the digestion and absorption process may also vary among different ingredients used in diets for pigs and consequently affect post-absorptive utilization of AA. Indeed, crystalline AA have been found to be more rapidly absorbed than protein bound AA [20], resulting in a lower utilization of AA. This was nicely illustrated by van den Borne et al. [21], who found a reduced utilization of AA in calves when the supplemented crystalline AA was given one time per day compared to four times per day. Currently, such differences are not taken into account, and there is a need to better understand how different commercial feed ingredients affect the absorption of AA and potentially the utilization of these AA. 

The present results showed that the absorption profile of AA was dependent on the feed ingredient with a greater postprandial concentration of total plasma AA and a greater peak in AA concentration observed for casein, hydrolyzed casein, and enzyme-treated soybean meal than for soybean meal, wheat, and hydrothermally-treated rapeseed meal. However, except for essential AA from soybean meal, the maximum absorption occurred 60 min after feeding. The major difference in absorption profiles were seen later on, although soybean meal and wheat had plasma concentrations close to the ones observed at 60 min. Therefore, the studied ingredients could merely be divided into slow and fast protein, but rather as protein sources with a short or long absorption. 

The concept of slow and fast proteins was first described in humans by Boirie et al. [2], who observed that absorption kinetics of AA between casein and whey protein showed different patterns, with casein having a lower initial concentration but a more prolonged appearance compared to whey protein. Another human study by Bos et al. [22] showed that soy protein isolate resulted in a faster AA appearance in plasma compared with casein. The lower concentration of AA in plasma of pigs fed soybean meal compared with casein is similar to observations made in other studies [23,24]. 

The rather similar absorption kinetics observed for the two animal protein products, casein and hydrolyzed casein in the present experiment was in contrast with previous human studies. Boirie et al. [2] described casein as a slow protein as it clots in the stomach and slows gastric emptying compared with hydrolyzed casein which is considered a fast protein, as it is already hydrolyzed [14]. Additionally, other human studies have shown that hydrolyzed casein results in a higher AA concentration in plasma and a faster appearance rate than casein [3,4]. However, in alignment of the present results, Horner et al. [5] found no significant differences between hydrolyzed casein and intact casein in obese humans. The higher accumulated total AA appearance in blood (area under the curve) following casein, but also hydrolyzed casein, during all time intervals is likely to be an effect of the higher digestibility of animal versus plant protein typically observed in pigs. 

Wheat can be categorized as a protein ingredient with prolonged absorption, but plasma AA concentration following the wheat diet increased very little 60 min after feeding. This may likely be related to the lower intake of protein from wheat compared with the other treatments. Anti-nutritional factors present in soybean meal, like phytate, non-starch polysaccharides, trypsin inhibitors, and lectins [25], have a negative effect on nutrient digestion [26]. Removal of anti-nutritional factors from the soybean meal may have contributed to the greater plasma concentration of AA following the enzyme-treated soybean meal diet compared to the soybean meal. Similarly, the hydrothermally-treated rapeseed meal was obtained by hydrothermal treatment of rapeseed meal in order to remove glucosinolates and improve AA availability. However, the hydrothermal treatment of the rapeseed meal, as such, could not be assessed in the present experiment because untreated rapeseed meal was not included in this study. The enzyme-treated soybean increased the overall plasma AA concentration compared with the soybean meal from which it was prepared, and the plasma AA concentration was significantly higher in pigs fed the enzyme-treated soybean at 60, 90, and 120 min after feeding. This difference is most likely related to an increase in digestibility caused by the enzyme treatment. However, the post-absorptive utilization of AA also depends on factors, such as a synchronized availability and concentration of carbohydrates/energy and AA [21]. The gross energy of the diets did only vary little, and consequently, the post-absorptive utilization of AA was likely affected more by the AA composition than by the energy concentration. 

In contrast to intact proteins, free AA are rapidly absorbed [21]. Although no statistical evaluation of the crystalline AA diet was performed in the present study due to feed refusal in most pigs, it is worth mentioning that raw data from a single pig successfully fed this treatment showed a very different pattern from the rest of the diets. Plasma AA concentrations peaked 30 min after feeding followed by a rapid decrease (data not shown).

Lysine showed a transient decrease in plasma concentration at 30 min after feeding and a low lysine plasma concentration in general (Appendix A), which may indicate an efficient extraction of this first limiting AA for protein synthesis. The concentration of methionine in plasma was very low for all treatments before feeding compared with other essential AA (Appendix A). Indeed, circulating essential AA in systemic blood is known to be low compared with non-essential AA [27,28]. The higher plasma methionine concentration after feeding casein and hydrolyzed casein illustrates the effect of the relative greater content of methionine in these ingredients. Similarly, the greater methionine concentration after feeding was higher for the hydrothermally-treated rapeseed meal compared with the soybean meal and enzyme-treated soybean meal, thus corresponding well with the hydrothermally-treated rapeseed meal containing relatively more methionine than the other plant based ingredients.

## 5. Conclusions

We conclude that the uptake and venous plasma clearance of amino acids in growing pigs was influenced by protein source, but the protein sources could not clearly be categorized as either “slow” or “fast” protein with regard to protein digestion and absorption of AA. However, soybean meal resulted in a prolonged and steady appearance of plasma AA compared to casein and hydrolyzed casein. The asynchronous absorption profiles may lead to imbalances in nutrient utilization. The AA absorption profile of an ingredient may be an important factor to assess in addition to other well-known contributors to the utilization of AA, such as the AA profile and the energy to protein ratio.

## Figures and Tables

**Figure 1 animals-11-01740-f001:**
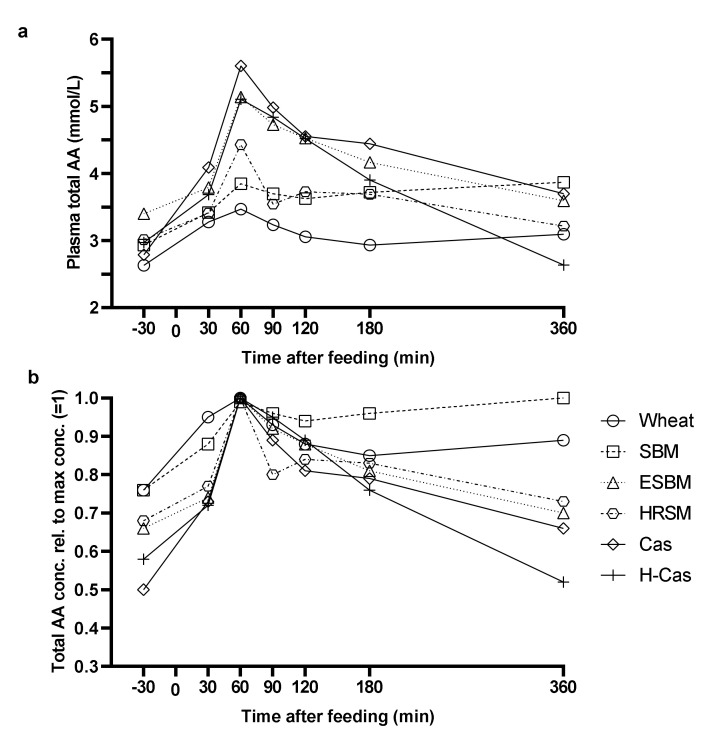
Postprandial total plasma AA concentration (**a**) and total plasma AA relative to the maximum AA concentration (=1) (**b**) of pigs fed diets containing wheat, soybean meal (SBM), enzyme-treated soybean meal (ESBM), hydrothermally-treated rapeseed meal (HRSM), casein (Cas), and hydrolyzed casein (H-Cas) as the only dietary protein sources at time 0. Values are least squares means with five replicates.

**Figure 2 animals-11-01740-f002:**
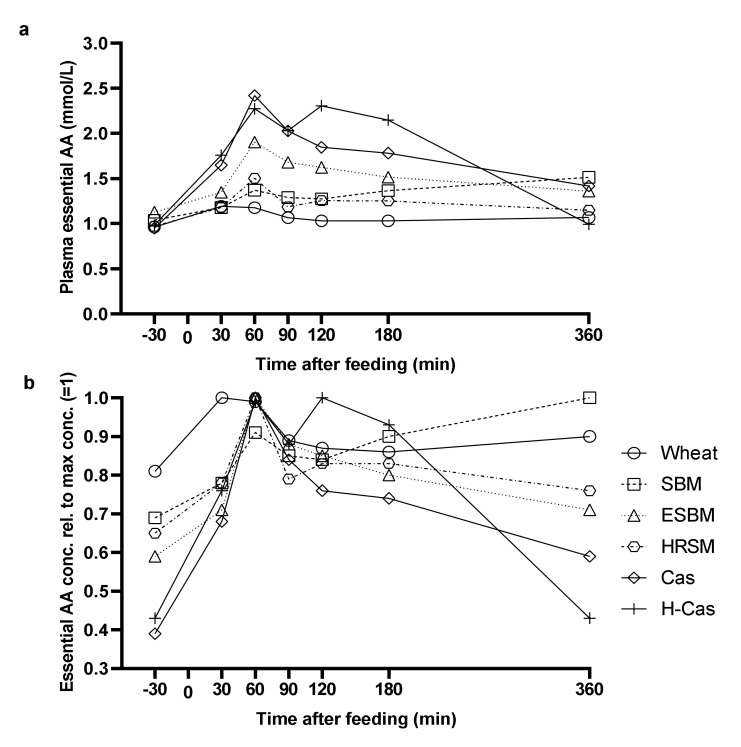
Postprandial essential amino acid concentration (**a**) and essential amino acid concentration relative to the maximum concentration (=1) (**b**) in plasma of pigs fed experimental diets containing wheat, soybean meal (SBM), enzyme-treated soybean meal (ESBM), hydrothermally-treated rapeseed meal (HRSM), casein (Cas), and hydrolyzed casein (H-Cas) as the only dietary protein source at time 0. Values are least squares means with five replicates.

**Figure 3 animals-11-01740-f003:**
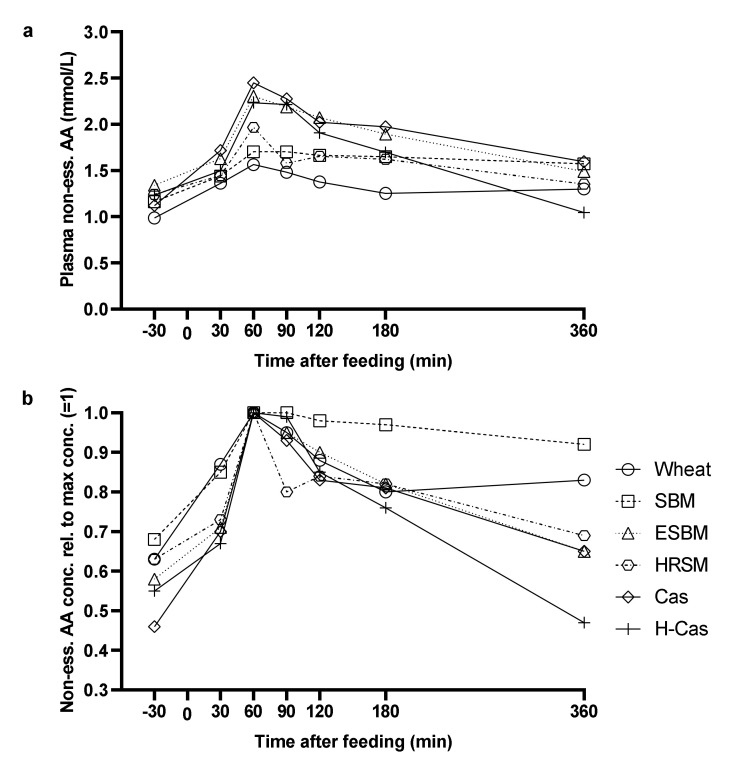
Postprandial non-essential amino acid concentration (**a**) and non-essential amino acid concentration relative to the maximum concentration (=1) (**b**) following each diet in plasma of growing pigs fed experimental diets containing wheat, soybean meal (SBM), enzyme-treated soybean meal (ESBM), hydrothermally-treated rapeseed meal (HRSM), casein (Cas), and hydrolyzed casein (H-Cas) as the only dietary protein sources. Values are least squares means with five replicates.

**Table 1 animals-11-01740-t001:** Composition of the seven experimental diets (%).

Ingredient	Wheat	SBM	ESBM	HRSM	Casein	H-Cas	C-AA
Wheat	100.00	-	-	-	-	-	-
SBM	-	40.70	-	-	-	-	-
ESBM	-	-	34.80	-	-	-	-
HRSM	-	-	-	61.50	-	-	-
Casein	-	-	-	-	24.30	-	-
H-Cas	-	-	-	-	-	24.30	-
C-AA ^1^	-	-	-	-	-	-	20.20
Cellulose fibre	-	2.74	2.74	2.74	2.74	2.74	2.74
Corn oil	-	4.56	4.56	4.56	4.56	4.56	4.56
Corn starch	-	38.20	44.10	17.40	54.60	54.60	58.70
Limestone	-	1.74	1.74	1.74	1.74	1.74	1.74
Min. vit. Premix ^2^	-	0.20	0.20	0.20	0.20	0.20	0.20
MCP ^3^	-	2.47	2.47	2.47	2.47	2.47	2.47
Salt	-	0.26	0.26	0.26	0.26	0.26	0.26
Sugar	-	9.13	9.13	9.13	9.13	9.13	9.13

SBM: soybean meal; ESBM: enzyme-treated soybean meal; HRSM: hydrothermally-treated rapeseed meal; H-Cas: hydrolyzed casein; C-AA: crystalline AA. ^1^ Composition of the crystalline AA is shown in Table 2. ^2^ Trouw Nutrition Denmark A/S. Content per g premix: 2500 IU vitamin A, 500 IU vitamin D_3_, 23.4 IU vitamin E, 0.6 mg vitamin K_3_, 0.6 mg vitamin B_1_, 1.2 mg vitamin B_2_, 3 mg d-panthotenicacid, 6.4 mg niacin, 0.060 mg biotin, 21.3 mg α-tocopherol, 0.006 mg vitamin B_12_, 0.6 mg vitamin B_6_, 50 mg Fe (Fe(II) sulphate), 41.3 mg Cu (Cu(II) sulphate), 50 mg Zn (Zn(II) oxide), 13.9 mg Mn (Mn(II) oxide), 0.076 mg KI, 0.075 mg Se (Se—selenite). ^3^ Mono-calcium phosphate.

**Table 2 animals-11-01740-t002:** Analyzed (except for the C-AA diet) crude protein (g/100 g DM) and amino acid (g/kg DM) content of protein ingredients.

	Wheat	SBM	ESBM	HRSM	Casein	H-Cas	C-AA
Crude protein	9.4	49.1	57.5	32.5	82.3	82.3	99.0
Essential AA							
Cys	2.4	8.5	8.9	8.4	3.8	3.8	-
His	2.4	14.3	15.8	9.7	28.6	28.6	28.6
Ile	3.7	27.8	30.6	15.5	49.4	49.4	49.4
Leu	7.0	42.4	48.4	25.7	93.5	93.5	93.5
Lys	3.1	33.2	34.1	20.3	77.5	77.5	77.5
Met	1.7	7.5	8.1	7.3	28.0	28.0	28.0
Phe	4.7	28.1	31.8	14.5	50.6	50.6	50.6
Thr	3.1	21.6	24.3	16.4	35.5	35.5	35.5
Trp	1.4	7.4	8.0	5.0	11.9	11.9	11.9
Val	4.8	27.5	30.9	19.9	65.1	65.1	65.1
Sum essential AA	34	218	241	143	444	444	440
Non-essential AA							
Ala	3.9	23.9	26.6	16.1	27.4	27.4	27.4
Arg	5.2	40.1	44.4	21.8	36.2	36.2	-
Asp	5.4	62.9	70.9	27.1	67.2	67.2	-
Glu	28.0	98.2	110.7	60.1	209.0	209.0	299.1
Gly	4.4	23.2	25.8	18.6	17.6	17.6	74.1
Pro	9.4	27.2	30.1	21.6	100.6	100.6	127.7
Ser	5.1	29.5	33.3	16.7	56.5	56.5	-
Sum non-essential AA	61	305	342	182	515	515	528
Sum AA	96	523	583	325	958	958	968

SBM = soybean meal; ESBM = enzyme-treated soybean meal; HRSM = hydrothermally-treated rapeseed meal; H-Cas = hydrolyzed casein; C-AA = crystalline AA.

**Table 3 animals-11-01740-t003:** Analyzed chemical composition (dry matter basis) of the experimental diets.

	N-Free ^1^	Wheat	SBM	ESBM	HRSM	Casein	H-Cas
Dry matter, g/100 g as-fed	92.2	87.5	91.2	92.7	90.0	92.0	92.8
Crude protein, g/100 g	0.4	10.9	20.3	21.7	21.8	22.1	19.1
Crude fat, g/100 g	5.2	2.4	4.1	4.7	4.6	4.0	4.0
Ash, g/100 g	5.1	1.8	5.2	5.9	7.0	6.2	5.8
Gross energy, MJ/kg	17.1	18.0	18.0	18.3	18.3	18.2	18.0
Essential AA, g/kg							
Cys	-	2.4	3.1	3.1	4.6	0.8	0.8
His	-	2.5	5.4	5.6	5.4	6.4	5.9
Ile	-	3.9	10.2	11.1	8.7	11.1	10.5
Leu	-	7.3	16.0	17.2	14.4	20.6	19.3
Lys	-	3.5	12.7	12.5	11.7	17.3	16.2
Met	-	1.7	2.8	2.9	4.1	6.2	5.8
Phe	-	4.8	10.5	11.3	8.1	11.2	10.5
Thr	-	3.2	8.1	8.6	9.2	7.9	7.4
Trp	-	1.5	2.7	2.8	2.9	2.8	2.4
Val	-	5.0	10.6	11.3	11.3	14.4	13.5
Sum essential AA	-	36	82	86	80	99	92
Non-essential AA, g/kg							
Ala	-	4.0	9.1	9.6	9.2	6.2	5.8
Arg	-	5.5	15.1	15.8	12.2	8.0	7.6
Asp	-	5.9	23.8	25.4	15.3	15.0	14.1
Glu	-	27.6	36.7	39.4	33.7	46.3	43.2
Gly	-	4.5	8.8	9.2	10.5	4.0	3.8
Pro	-	9.6	10.3	11.0	12.3	22.7	21.1
Ser	-	5.2	11.0	11.8	9.3	12.5	11.7
Sum non-essential AA	-	62	115	122	103	115	107
Sum AA		98	197	209	183	213	200

N-free = Nitrogen-free ingredients; SBM = soybean meal; ESBM = enzyme-treated soybean meal; HRSM = hydrothermally-treated rapeseed meal; H-Cas = hydrolyzed casein. ^1^ AA were analyzed but were below detection limit.

**Table 4 animals-11-01740-t004:** Quantitative postprandial concentration expressed as area under the curve (min × mM) of AA in plasma of pigs fed experimental diets containing one protein source.

Period	Diets							
	Wheat	SBM	ESBM	HRSM	Casein	H-Cas	SEM	*p*-Value ^1^
0–30 min	95 ^b^	98 ^a,b^	101 ^a,b^	96 ^b^	115 ^a^	103 ^a,b^	5.6	0.04
0–60 min	198 ^b^	210 ^a,b^	236 ^a,b^	205 ^b^	268 ^a^	239 ^a,b^	16.8	0.009
0–90 min	301 ^b^	324 ^a,b^	385 ^a,b^	317 ^b^	425 ^a^	389 ^a,b^	27.4	0.003
0–120 min	398 ^b^	434 ^a,b^	524 ^a,b^	426 ^b^	565 ^a^	532 ^a,b^	36.7	0.002
0–180 min	582 ^b^	655 ^a^	786 ^a^	649 ^a,b^	830 ^a^	832 ^a^	55.4	0.002
0–360 min	1137 ^b^	1340 ^a b^	1485 ^a,b^	1283 ^a,b^	1544 ^a^	1519 ^a,b^	106.4	0.02

SBM = soybean meal; ESBM = enzyme-treated soybean meal; HRSM = hydrothermally-treated rapeseed meal; H-Cas = hydrolyzed casein. ^1^ Values within the row with different superscript letters (^a, b^) differ significantly (*p* < 0.05; Tukey-Kramer adjusted).

## Data Availability

Data is contained within the article.

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
