# Peer review of "Amino Acid Absorption Profiles in Growing Pigs Fed Different Protein Sources"

_animals, 2021, doi:10.3390/ani11061740_

Round 1

Reviewer 1 Report

General comments

The present study negotiates an interesting topic on the amino acid absorption profiles in pigs fed seven kinds of representative protein sources, which will benefit to optimize the formulation of diets for pigs. The experiment was designed well with appropriate methods to verify the hypothesis. However, there are several points that need to be addressed and explained by the authors before it can be published. Specific comments are provided below.

Line 76: Why did the authors choose wheat as a source of protein in the present study?

L116: “25-35 g/kg BW0.75”. How to choose between 25 g or 35 g?

L127-128: Please explain what is the meaning of “the feed was removed 15 minutes later”? What if the pigs did not finish eating, as there might be some pigs not eating quickly? For example, the pigs eating the C-AA diet.

L129: How is the time 0 of continuous blood collection determined?Time 0 is set according to the initiation of the morning meal or the end of the morning meal. If Time 0 is set according to the initiation of the morning meal, whether the results would be influenced by the length of feeding time? Please explain.

L132: “-80” >> “-80”.

Line 197-198: The author may want to explain how was the “essential amino acid concentration relative to the maximum concentration” calculated.

L292-293, 295-299: The authors referred some data not shown. The author may want to add them to the article or supplementary files, because the article is incomplete or unconvincing without these parts.

Why did the authors provide the absorption profiles of Lys and Met in “Result” part?

The significant differences in the areas under the curve of AA in plasma should be discussed in the “Discussion” part.

Line 334: Eur. J. Nutr >> Eur. J. Nutr.

Line 389: Nutr. Res. Rev. >>Nutr. Res. Rev. Please check the reference format again.

Reviewer 2 Report

Interesting work, from the area of basic research, comparing the rate and intensity of absorption of amino acids from different sources based on their concentration in the blood of growing pigs.

Notes to work:

I propose to change the title of the work. Studies were carried out on young pigs, hence the suggestion to write "young pigs" or "growing pigs" in the title, the animals had a body weight of about 18kg in the experiment.

What was the criterion for choosing protein sources in the experiment?

In the Authors' view, is the relatively constant and longer-lasting absorption of amino acids from SBM beneficial to the pig organism and confirms the popularity of SBM as a source of protein in pig feed?

Did the Authors analyze the reason why pigs refused to eat the C-AA diet?

Were the animals in the studies healthy (digestive health) and under the control of a veterinarian?

Author Response

Please see the attached file including the response to both reviewers comments (same file as uploaded to reviewer 1)

Round 2

Reviewer 1 Report

All the comments of this reviewer have been addressed.